# PRECONDITIONED NORMS: A UNIFIED FRAMEWORK FOR STEEPEST DESCENT, QUASI-NEWTON AND ADAPTIVE METHODS

## ABSTRACT

Optimization lies at the core of modern deep learning, yet existing methods often face a fundamental trade-off between adapting to problem geometry and leveraging curvature utilization. Steepest descent algorithms adapt to different geometries through norm choices but remain strictly first-order, whereas quasi-Newton and adaptive optimizers incorporate curvature information but are restricted to Frobenius geometry, limiting their applicability across diverse architectures. In this work, we propose a unified framework generalizing steepest descent, quasi-Newton methods, and adaptive methods through the novel notion of preconditioned matrix norms. This abstraction reveals that widely used optimizers such as SGD and Adam, as well as more advanced approaches like Muon and KL-Shampoo, and recent hybrids including SOAP and SPlus, all emerge as special cases of the same principle. Within this framework, we provide the first systematic treatment of affine and scale invariance in the matrix-parameterized setting, establishing necessary and sufficient conditions under generalized norms. Building on this foundation, we introduce two new methods, `MuAdam` and `MuAdam-SANIA`, which combine the spectral geometry of Muon with Adam-style preconditioning. Our experiments demonstrate that these optimizers are competitive with, and in some cases outperform, existing state-of-the-art methods. Our code is available at https://anonymous.4open.science/r/LIB-2D05

## 1 INTRODUCTION

Optimization lies at the heart of modern machine learning (Bottou, 2010; Goodfellow et al., 2016; Team et al., 2025), including today's most prominent systems such as Large Language Models (Vaswani et al., 2017; Brown et al., 2020; Hernández-Cano et al., 2025) and generative AI (Goodfellow et al., 2014; Rombach et al., 2022). Improvements in optimization efficiency or stability directly translate into faster training, reduced computational costs, and ultimately better-performing models (Kingma & Ba, 2014). Training a model amounts to adjusting its parameters $W$ to minimize a loss function $\mathcal{L}(W)$, which measures the discrepancy between predictions and data. While this formulation has long been fundamental to learning theory and practice, deep learning introduces new challenges: datasets are very large (Dean et al., 2012; Tian et al., 2025), parameter spaces are extremely high-dimensional (Team et al., 2025; Hernández-Cano et al., 2025), and the loss landscapes are highly non-convex (Choromanska et al., 2015; Chen et al., 2025).

As computing the full gradient is computationally prohibitive, the standard approach to these challenges is *stochastic optimization* (Robbins & Monro, 1951; Rumelhart et al., 1986; Tian et al., 2023). Instead of computing the full loss, training samples $\xi \sim \mathcal{D}$ are drawn from the underlying data distribution, and the expected loss

$$\mathcal{L}(W) = \mathbb{E}_{\xi \sim \mathcal{D}} \mathcal{L}(W; \xi)$$

is minimized using stochastic gradients $G_t = \nabla \mathcal{L}(W_t; \xi_t)$ computed on samples or minibatches (Bottou, 2010; Shalev-Shwartz & Ben-David, 2014; Ward, 2022). The model parameters are then updated iteratively (Rumelhart et al., 1986; Bernstein & Newhouse, 2024b):

$$W_{t+1} = W_t - \alpha_t \Delta W_t, \tag{1}$$

where $\alpha_t$ is the learning rate and the update rule $\Delta W_t$ depends on the chosen optimization method. For example, Stochastic Gradient Descent (SGD) corresponds to $\Delta W_t = G_t$ (Rumelhart et al., 1986). This general template encompasses most practical algorithms, ranging from Muon (Jordan et al., 2024) to quasi-Newton methods (Gupta et al., 2018) and adaptive optimizers (Kingma & Ba, 2014).

Traditional optimization approaches typically vectorize all parameters, treating them as elements in the space $\mathbb{R}^{mn}$ Rumelhart et al. (1986). Recent work emphasizes that in practice, parameters $W$ possess natural *matrix structure*: $W \in \mathbb{R}^{m \times n}$ (e.g., the weights of linear or convolutional layers), which can be exploited to achieve faster and more robust convergence (Gupta et al., 2018; Goldfarb et al., 2020; Bernstein & Newhouse, 2024a; Jordan et al., 2024; Vyas et al., 2024; Pethick et al., 2025; Riabinin et al., 2025).

A useful perspective on update rules is the classical *steepest descent* interpretation (Bernstein & Newhouse, 2024b; Pethick et al., 2025): the update direction corresponds to the direction of steepest loss decrease under a chosen norm, where the choice of norm defines the optimization geometry. For vector-valued parameters, $\ell_2$ steepest descent recovers normalized SGD (Hazan et al., 2015), while $\ell_\infty$ steepest descent yields SignSGD (Bernstein et al., 2018). Recently, for matrix-valued parameters, Muon-like algorithms have employed the spectral (singular value) matrix norm (Jordan et al., 2024; Pethick et al., 2025; Riabinin et al., 2025), which explicitly leverages the structural properties of weight matrices. The steepest descent viewpoint therefore unifies many modern optimizers. However, these algorithms do not incorporate second-order curvature information, which naturally motivates quasi-Newton-based techniques (Liu & Nocedal, 1989; Gupta et al., 2018).

Quasi-Newton methods approximate curvature by introducing a positive definite matrix $H_t \succeq 0$ that estimates the Hessian $\nabla^2 \mathcal{L}(W_t)$ (Goldfarb et al., 2020; Gao et al., 2024). In the vector case, they produce preconditioned updates of the form $\Delta W_t = H_t^{-1} G_t$ (Davidon, 1959; Broyden, 1970; Fletcher, 1970; Goldfarb, 1970; Shanno, 1970; Broyden, 1967; Liu & Nocedal, 1989). Beyond faster convergence, quasi-Newton methods are popular due to their natural geometric properties—they preserve the sequence of iterates regardless of function scaling or choice of basis (the so-called *affine invariance* (Nesterov & Nemirovski, 1994; Nesterov et al., 2018; d'Aspremont et al., 2018)), facilitating implementation and hyperparameter tuning. In the matrix case, quasi-Newton updates naturally generalize to:

$$\Delta W_t = (H_t^L)^{-1} \cdot G_t \cdot (H_t^R)^{-1}, \tag{2}$$

where $H_t^L$ and $H_t^R$ approximate left and right curvature factors, and their Kronecker product acts as a surrogate Hessian $H_t$. This principle underlies widely used methods such as K-FAC (Martens & Grosse, 2015), Shampoo (Gupta et al., 2018), and SOAP (Vyas et al., 2024), which exploit layer-wise gradient covariances for computational scalability.

Another important approach to Hessian estimation involves element-wise preconditioners such as AdaGrad (Duchi et al., 2011), RMSProp (Tieleman, 2012), and Adam (Kingma & Ba, 2014), which, in the vector case, can be interpreted as quasi-Newton updates with diagonal $H_t$ and possess the property of *scale invariance* (Abdukhakimov et al., 2023; Choudhury et al., 2024). By contrast, adaptive methods in the matrix domain use the Hadamard product, yielding updates of the form

$$\Delta W_t = V_t^{\circ -1} \odot G_t, \tag{3}$$

where $V_t$ is a matrix with positive elements (scaling factors), $\odot$ denotes the Hadamard product, and $V_t^{\circ -1}$ denotes the element-wise inverse. These are also referred to as *adaptive methods*, and due to their element-wise nature, they enjoy scale invariance rather than affine invariance.

Together, quasi-Newton and element-wise preconditioned methods constitute the two main paradigms for incorporating curvature information into deep learning optimization. While these approaches can achieve important geometric properties such as affine or scale invariance, they remain fundamentally constrained to the Frobenius norm, limiting their ability to capture more complex geometric structures. In contrast, steepest descent methods offer flexibility in norm selection but inherently lack these crucial invariance properties, thereby restricting their effectiveness when optimizing across the diverse structural landscapes of modern deep learning architectures. These limitations motivate the central question of our work:

*Can optimization algorithms inherit both the geometric adaptability of steepest descent and the curvature-awareness of quasi-Newton and adaptive approaches?*

Our work provides a positive answer to this question, with the following main contributions:

1. **Unification of optimization methods through generalized norms.** We develop a comprehensive framework based on generalized norms (Definitions 2.1 and 2.2) that reveals fundamental connections between seemingly disparate optimization approaches. Our framework demonstrates that classical steepest descent, quasi-Newton methods, and adaptive element-wise algorithms are special cases of the same underlying principle, while also providing a principled interpretation of recently proposed methods like SOAP (Vyas et al., 2024) and SPlus (Frans et al., 2025).

2. **Systematic optimizer design methodology.** We derive the steepest descent update for arbitrary generalized norms (Theorem 2.3), establishing a principled framework for designing new optimizers. Our approach enables the creation of novel algorithms by combining different curvature approximations with various descent geometries, as demonstrated with our newly proposed optimizer (Algorithm 1).

3. **First systematic analysis of invariance properties in matrix-parametrized optimization.** We provide the first comprehensive study of affine and scale invariance in the matrix-parametrized setting, deriving explicit necessary and sufficient conditions within our framework (Theorem 3.1).

4. **Empirical validation of proposed methods and invariance properties.** We demonstrate the practical effectiveness of our framework through extensive experiments, showcasing the performance advantages of our proposed optimizers and empirically verifying their theoretical invariance properties (Section 4).

The remainder of the paper is structured as follows. In the next section, we define technical notation and discuss related work. In Section 2, we present our unifying framework, where we introduce generalized norms and their connections to the literature, and derive the corresponding update steps. In Section 3, we establish necessary and sufficient conditions for affine and scale invariance in the matrix-parametrized case. Finally, in Section 4, we present our numerical evaluation.

## 1.1 PRELIMINARIES

The steepest descent principle provides one of the most classical formulations of first-order optimization (Bernstein & Newhouse, 2024a;b). In this framework, each update direction is defined as the solution of a norm-constrained quadratic model of the loss. More recently, this perspective has been revisited in deep learning, where the update step is written in terms of a *Linear Minimization Oracle* (LMO) (Pethick et al., 2025). Concretely, given the matrix $G_t \in \mathbb{R}^{m \times n}$, the LMO is defined as

$$\mathrm{lmo}(G_t) \in \operatorname*{argmax}_{T \in \mathbb{R}^{m \times n} : \|T\| \leq \rho} \langle G_t, T \rangle, \tag{4}$$

where $\|\cdot\|$ is a matrix norm, $\rho > 0$ is a scaling parameter, and $\langle \cdot, \cdot \rangle$ denotes the Frobenius inner product. The corresponding steepest descent update then takes the form

$$\Delta W_t = \mathrm{lmo}(G_t). \tag{5}$$

A general and widely studied family is given by the $\alpha \to \beta$ operator norms:

$$\|G\|_{\alpha \to \beta} = \sup_{\|x\|_\alpha \leq 1} \|Gx\|_\beta,$$

where $\|\cdot\|_\alpha$ and $\|\cdot\|_\beta$ are vector norms. A first important case uses the root-mean-square norm, defined for $x \in \mathbb{R}^d$ as $\|x\|_{\mathrm{RMS}} := \dim(x)^{-\frac{1}{2}} \|x\|_2$. When $\alpha = \beta = \mathrm{RMS}$, the resulting operator norm coincides with the spectral norm and the corresponding steepest descent method is known as Muon (Jordan et al., 2024). Here, the core step is the spectral LMO, which for gradients with singular value decomposition $G_t = U_t \Sigma_t V_t^\top$ returns $U_t V_t^\top$. Muon avoids computing the explicit SVD decomposition via Newton–Schulz iteration (Bernstein & Newhouse, 2024b), an efficient procedure based only on matrix multiplications, to approximate this projection onto the spectral geometry. Beyond the spectral case, other choices of $\alpha$ and $\beta$ yield different but related update rules. Two further notable instances are column-normalized and row-normalized steps using the $1 \to \mathrm{RMS}$ and $\mathrm{RMS} \to \infty$ norms (Pethick et al., 2025).

These constructions illustrate how steepest descent can naturally incorporate matrix geometry, but they remain fundamentally tied to first-order information, as no curvature or higher-order structure of the loss $\mathcal{L}$ is explicitly taken into account. This limitation motivates the quasi-Newton and adaptive approaches discussed in the next section.

## 1.2 QUASI-NEWTON AND ADAPTIVE METHODS

In the matrix setting, quasi-Newton updates are usually expressed through two sided preconditioning,

$$\Delta W_t = (L_t^\top L_t)^{-1} G_t (R_t^\top R_t)^{-1}, \tag{6}$$

where, in contrast to (2), we use matrices $L_t$ and $R_t$, chosen such that $L_t^\top L_t := H_t^L$ and $R_t^\top R_t := H_t^R$. This parametrization is purely for convenience (see Definition 2.1 and Theorem 2.3).

Several choices have been studied in the literature:

$$H_t^L = \sum_{s=0}^{t-1} G_s G_s^\top, \qquad H_t^R = \sum_{s=0}^{t-1} G_s^\top G_s, \qquad \text{(Shampoo (Gupta et al., 2018))}$$

$$H_t^L = (1-\beta)H_{t-1}^L + \beta G_s G_s^\top, \quad H_t^R = (1-\beta)H_{t-1}^R + \beta G_s^\top G_s, \quad \text{(SOAP (Vyas et al., 2024))}$$

with $\beta \in (0,1)$ controlling exponential averaging. These constructions approximately factorize the Hessian via the Kronecker structure $H_t = H_t^R \otimes H_t^L$ and are representative of the widely used *Kronecker-factored preconditioning* family (Martens & Grosse, 2015; Zhang et al., 2025).

Beyond such Kronecker-based methods, adaptive optimizers construct element-wise diagonal preconditioners. A general update can be written as

$$\Delta W_t = (D_t \odot D_t)^{\circ -1} \odot G_t, \tag{7}$$

where $D_t \in \mathbb{R}^{m \times n}$ stores positive coordinate-wise statistics, $\circ - 1$ denotes element-wise inversion, and $\odot$ is the Hadamard product. Analogously to the quasi-Newton case here we use $D_t \odot D_t := V_t$ instead of $V_t$ as in (3) for convenience. Notable examples of adaptive preconditioners include:

$$V_t = \left( \sum_{s=0}^{t-1} G_s \odot G_s \right)^{\circ \frac{1}{2}}, \qquad \text{(AdaGrad (Duchi et al., 2011))}$$

$$V_t = \left( \beta V_{t-1} + (1-\beta)(G_t \odot G_t) \right)^{\circ \frac{1}{2}}. \qquad \text{(Adam (Kingma & Ba, 2014))}$$

Thus, while quasi-Newton methods exploit Kronecker-factored approximations of curvature through $H_t^L, H_t^R$, adaptive methods rely on element-wise preconditioners $V_t$ derived from gradient magnitudes. These formulations illustrate the range of preconditioning strategies used in deep learning, and serve as key reference points for the unified framework proposed in this work.

## 2 NOVEL FRAMEWORK

Our goal is to unify steepest descent, quasi-Newton, and adaptive methods under a single geometric framework. The central idea is to define new families of matrix norms that encode preconditioning directly, and then to characterize the associated LMOs, which determine the update steps.

We first introduce two classes of norms that generalize both quasi-Newton and adaptive updates.

**Definition 2.1.** For any matrix $G \in \mathbb{R}^{m \times n}$, positive definite matrices $L \in \mathbb{R}^{m \times m}$, $R \in \mathbb{R}^{n \times n}$, and a base matrix norm $\|\cdot\|$, we call a $(L, R)$-*preconditioned matrix norm*,

$$\|G\|_{L,R,\|\cdot\|} \overset{\text{def}}{=} \|L \cdot G \cdot R\|.$$

This general norm includes as a special case Kronecker-factored methods when the base norm is Frobenius ($\|\cdot\|_{L,R,\|\cdot\|_F}$), which leads to the quasi-Newton update (6) from the LMO step (4) Li (2024).

**Definition 2.2.** For any matrix $G \in \mathbb{R}^{m \times n}$, diagonal positive matrix $D \in \mathbb{R}^{m \times n}$, and any matrix norm $\|\cdot\|$, we call $D$-*preconditioned matrix norm*,

$$\|G\|_{D,\|\cdot\|} = \|D \odot G\|.$$

This formulation includes adaptive optimizers with Frobenius base norm.

Together, these definitions capture the two major strands of preconditioning in deep learning: Kronecker-based curvature approximations via $(L, R)$ and coordinate-wise scaling via $D$. Note, that if all preconditioners are chosen as identity matrices, the framework reduces to the classical steepest descent updates, depending on the base norm $\|\cdot\|$. Thus, many classical algorithms emerge as special cases of the generalized construction.

Linear minimization oracles associated with these norms have a simple structure: they reduce to the LMO of the underlying base norm in a transformed gradient space. To clarify technical details, let's explicitly denote the choice of the base norm in the LMOs an $\text{lmo}_{\|\cdot\|}$.

**Theorem 2.3.** *The linear minimization oracles for $(L, R)$-norm and $D$-norm can be expressed as*[1]

$$lmo_{L,R,\|\cdot\|}(G) = L^{-1} lmo_{\|\cdot\|}(L^{-T} G R^{-T}) R^{-1},$$

$$lmo_{D,\|\cdot\|}(G) = D^{\circ-1} \odot lmo_{\|\cdot\|}(D^{\circ-1} \odot G).$$

Theorem 2.3 highlights that in the preconditioned setting the LMO acts within a transformed gradient space. For the $(L, R)$-norms, the gradient is mapped to the transformed space $L^{-T} G R^{-T}$ where the base LMO is applied, and the result is mapped back to the original space by $L^{-1}$ and $R^{-1}$. This general characterization of LMO enables unification of seemingly different approaches – norm-constrained steepest descent, Kronecker-factored quasi-Newton methods, adaptive optimizers, and recent hybrids all emerge as special cases. We summarized this observations in Table 1.

Table 1: Popular optimization methods and their parameterization within the unified framework. Sign "$-$" indicates that the corresponding parameter is not used in the method. EMA indicates that the exponential moving average of the corresponding quantity is utilized.

| | Method | $L_t$ | $R_t$ | $D_t$ | Base Norm |
|---|---|---|---|---|---|
| **Norm-based** | Normalized SGD (Hazan et al., 2015) | $-$ | $-$ | $-$ | Frobenius |
| | SignSGD (Bernstein et al., 2018) | $-$ | $-$ | $-$ | $\ell_1 \to \ell_\infty$ |
| | Muon (Jordan et al., 2024) | $-$ | $-$ | $-$ | Spectral |
| | Scion (Pethick et al., 2025) | $-$ | $-$ | $-$ | matrices: Spectral
vectors: $\ell_\infty$ |
| **Quasi-Newton** | K-FAC (Martens & Grosse, 2015) | $(\mathbb{E}[GG^\top])^{1/8}$ | $(\mathbb{E}[G^\top G])^{1/8}$ | $-$ | Frobenius |
| | Shampoo (Gupta et al., 2018) | $(\sum_{s=1}^{t-1} G_s G_s^\top)^{1/8}$ | $(\sum_{s=1}^{t-1} G_s^\top G_s)^{1/8}$ | $-$ | Frobenius |
| | One-sided Shampoo (Xie et al., 2025) | $(\sum_{s=1}^{t-1} G_s G_s^\top)^{1/4}$ | $I$ | $-$ | Frobenius |
| | KL-Shampoo (Lin et al., 2025) | $(\text{EMA}[G_t[R_t^T R_t]^{-1} G_t^\top])^{1/8}$ | $(\text{EMA}[G_t^\top [L_t^T L_t]^{-1} G_t])^{1/8}$ | $-$ | Frobenius |
| **Adaptive** | AdaGrad (Duchi et al., 2011) | $-$ | $-$ | $(\sum_{s=1}^{t-1} G_s \odot G_s)^{1/4}$ | Frobenius |
| | Adam (Kingma & Ba, 2014) | $-$ | $-$ | $(\text{EMA}[G_t \odot G_t])^{1/4}$ | Frobenius |
| | MADGRAD (Defazio & Jelassi, 2022) | $-$ | $-$ | $(\text{EMA}[G_t \odot G_t])^{1/6}$ | Frobenius |
| | Adam-SANIA (Abdukhakimov et al., 2023) | $-$ | $-$ | $(\text{EMA}[G_t \odot G_t])^{1/2}$ | Frobenius |
| **Hybrid** | SOAP (Vyas et al., 2024) | $Q_L$ [1] | $Q_R$ [1] | $-$ | $\|\cdot\|_{D=\text{Adam}, \|\cdot\|_F}$ [2] |
| | SPlus (Frans et al., 2025) | $Q_L$ [1] | $Q_R$ [1] | $-$ | $\ell_1 \to \ell_\infty$ |
| | MuAdam (Algorithm 1 with $p = 1/4$) | $-$ | $-$ | $(\text{EMA}[G_t \odot G_t])^{1/4}$ | Spectral |
| | MuAdam-SANIA (Algorithm 1 with $p = 1/2$) | $-$ | $-$ | $(\text{EMA}[G_t \odot G_t])^{1/2}$ | Spectral |

[1] $Q_L, Q_R$ are eigenbasis matrices from Shampoo's preconditioners $\sum_{s=1}^{t-1} G_s G_s^\top$ and $\sum_{s=1}^{t-1} G_s^\top G_s$ respectively.
[2] $\|\cdot\|_{D=\text{Adam}, \|\cdot\|_F}$ denotes the $D$-norm with Adam diagonal preconditioning and Frobenius norm.

Among the methods listed in Table 1, hybrid methods SOAP (Vyas et al., 2024) and SPlus (Frans et al., 2025) deserve special attention. Both explicitly combine the geometry of quasi-Newton style preconditioning with a linear minimization oracle step: in SOAP, Shampoo's Kronecker preconditioners (Gupta et al., 2018) are coupled with Adam-style diagonal adaptation (Kingma & Ba, 2014) performed in the preconditioned eigenbasis, while in SPlus, the same Kronecker structure is combined with a sign-based LMO (Bernstein et al., 2018). These optimizers exemplify precisely the type of integration that our theory highlights: the LMO does not replace the preconditioner but interacts with it in a structured way. Empirically, both SOAP and SPlus have shown strong performance on large-scale deep learning benchmarks (Semenov et al., 2025; Wen et al., 2025), improving efficiency and robustness compared to their constituent parts. They confirm that blending norm-based updates with second-order preconditioning yields practical benefits, such as stability at high learning rates (Frans et al., 2025) and faster convergence with reduced hyperparameter tuning (Vyas et al., 2024).

---

[1] For an invertible matrix $M$, we use shorthand notation for the inverse transpose as $M^{-T}$.

Building on this idea, we introduce two new optimizers: `MuAdam` and `MuAdam-SANIA` (Algorithm 1). Both use Muon's spectral LMO (Jordan et al., 2024) in conjunction with Adam-style (Kingma & Ba, 2014) or SANIA (Abdukhakimov et al., 2023) preconditioners. While `MuAdam` can be seen as a practical analogue of Adam within a spectral geometry, `MuAdam-SANIA` extends this construction with the scale-invariant SANIA update, providing robustness to coordinate-wise rescaling. `MuAdam` can be viewed as a practical optimizer in the same spirit as Adam, while `MuAdam-SANIA` inherits scale-invariance from its preconditioner, offering robustness to coordinate-wise rescaling.

---

**Algorithm 1** `MuAdam` and `MuAdam-SANIA`

---

1: **Parameters:** step size $\gamma_t$, Adam coefficients $\beta_1, \beta_2 \in (0, 1)$, stability constant $\varepsilon > 0$.
2: **Initialization:** $M_{-1} = V_{-1} = 0$.
3: **for** $t = 0, 1, 2, \ldots$ **do**
4:     **Gradient estimation:** obtain stochastic gradient $G_t = \nabla \mathcal{L}(W_t; \xi_t)$.
5:     **Moment and precondition updates:**                   (Adam (Kingma & Ba, 2014))

$$M_t = \beta_1 M_{t-1} + (1 - \beta_1)G_t, \quad \hat{M}_t = \frac{M_t}{1 - \beta_1^{t+1}},$$

$$V_t = \beta_2 V_{t-1} + (1 - \beta_2)(G_t \odot G_t), \quad \hat{V}_t = \frac{V_t}{1 - \beta_2^{t+1}}.$$

6:     **First preconditioning:**

$$N_t = \frac{\hat{M}_t}{\hat{V}_t^{\circ p} + \varepsilon}, \quad \text{where } p = \begin{cases} 1/4, & \text{for } \texttt{MuAdam}, \\ 1/2, & \text{for } \texttt{MuAdam-SANIA}. \end{cases}$$

7:     **Spectral norm based LMO step:**                     (Muon (Jordan et al., 2024))

$$N_t' = \text{Newton–Schulz}(N_t).$$

8:     **Second preconditioning:**

$$N_t'' = \frac{N_t'}{\hat{V}_t^{\circ p} + \varepsilon}.$$

9:     **Update:** $W_{t+1} = W_t - \gamma_t \cdot N_t''$.
10: **end for**

---

In Algorithm 1 we incorporate the momentum term $M_t$, which is standard in modern optimization (Polyak, 1964; Kingma & Ba, 2014). Furthermore, the appearance of two applications of the element-wise preconditioner $D_t = \hat{V}_t^{\circ p} + \varepsilon$ follows directly from Theorem 2.3, which describes how precondition matrices interact with the LMO of the base norm.

Taken together, SOAP (Vyas et al., 2024), SPlus (Frans et al., 2025), and proposed `MuAdam` and `MuAdam-SANIA` (Algorithm 1) illustrate the potential of systematic combinations of norm-based LMOs with quasi-Newton or adaptive preconditioners. Our framework highlights that this design space is broad but still under-explored, suggesting a promising direction for future research.

## 3 GEOMETRIC PROPERTIES

As we mentioned earlier, one of the important properties of optimization algorithms is *affine* and *scale invariance* (Abdukhakimov et al., 2023). Informally, an algorithm is affine invariant if its behavior is unaffected by a linear reparametrization of the parameters (e.g., changes to the coordinate basis), whereas scale invariance refers to invariance under coordinate-wise rescaling. These properties are desirable because they facilitate the implementation process, ensure that optimization dynamics reflect only geometry of the objective, and can lead to improvement in accuracy (Yen et al., 2024). In Appendix A we provide a detailed discussion of affine and scale invariance for common optimizers.

**Invariances for vectors.** For the vector-valued losses, affine invariance has been considered only for transformations of the form $\mathcal{L}_{\text{new}}(w) = \mathcal{L}(Aw)$ with a non-degenerate matrix $A$ (Abdukhakimov et al., 2023; Choudhury et al., 2024). This covers both full linear reparametrizations and, as a special

case of diagonal $A$, *scale invariance*. Formally, we say that an optimization algorithm is scale/affine *invariant* if the iterate sequences of algorithm coincide on $\mathcal{L}$ and its reparametrized version $\mathcal{L}_{\text{new}}$:

$$\mathcal{L}(W_t) = \mathcal{L}_{\text{new}}(W_t^{\text{new}}) \quad \text{for all iterations } t,$$

where $\{W_t\}$ and $\{W_t^{\text{new}}\}$ denote the iterates produced for $\mathcal{L}$ and $\mathcal{L}_{\text{new}}$, respectively.

**Example.** Consider a linear model $\mathcal{L}(w) = (\langle x, w \rangle - y)^2$ with vector parameters $w$. If the inputs are linearly transformed, $x \mapsto A^T x$, then the corresponding reparametrized loss is

$$\mathcal{L}_{\text{new}}(w) = (\langle A^T x, w \rangle - y)^2 = (\langle x, Aw \rangle - y)^2 = \mathcal{L}(Aw).$$

An affine invariant optimization algorithm should then generate identical optimization dynamics under such a change of variables. If $A$ is diagonal, this reduces to coordinate-wise scaling.

**Invariances for matrices.** In this work, we extend these notions for the first time to the case of matrix-valued parameters, which are ubiquitous in modern deep learning. We introduce affine invariance in this setting by defining the reparametrized function as

$$\mathcal{L}_{\text{new}}(W) = \mathcal{L}(A_L W A_R),$$

where $A_L$ and $A_R$ are non-degenerate matrices. This generalization is consistent with the classical vector definition but now naturally decomposes into two sides: left multiplication reflects a change of basis or rescaling of input features, while right multiplication corresponds to transformations of the output representation. Analogously, scale invariance in the matrix case takes the form of *element-wise reparametrizations* through the Hadamard product,

$$\mathcal{L}_{\text{new}}(W) = \mathcal{L}(A \odot W),$$

where $A$ is now a positive scaling matrix. Thus, while in the vector case affine and scale invariance reduce to linear and diagonal transformations, in the matrix case they naturally separate into two distinct but related notions: left/right affine transformations and element-wise scaling.

We are going to formalize the conditions under which the LMO step (5) with norms from Definitions 2.1 and 2.2 exhibits affine and scale invariance in the matrix setting. In both cases, the result provides a necessary and sufficient characterization, directly expressed in terms of the transformation rules for the preconditioners. Result for general norm is presented in Theorem B.1 in Appendix B.

**Theorem 3.1.** *Let the LMO step* (5) *be implemented with the preconditioned norms (as Theorem 2.3) with a base norm $\|\cdot\|$ such that the solution is unique. Then the conditions for the invariances are:*

*Affine invariance: For the $(L, R, \|\cdot\|)$-norm (Definition 2.1), the step is affine invariant with respect to transformations $\mathcal{L}_{new}(W) = \mathcal{L}(A_L W A_R)$ if and only if the left and right preconditioners $(L^{\mathcal{L}}, R^{\mathcal{L}})$ for the loss $\mathcal{L}$ and $(L^{\mathcal{L}_{new}}, R^{\mathcal{L}_{new}})$ for the loss $\mathcal{L}_{new}$ satisfy*

$$L^{\mathcal{L}_{new}} = L^{\mathcal{L}} A_L, \qquad R^{\mathcal{L}_{new}} = A_R R^{\mathcal{L}},$$

*Scale invariance: For the $D$-norm (Definition 2.2), the step is scale invariant with respect to element-wise rescaling $\mathcal{L}_{new}(W) = \mathcal{L}(A \odot W)$ if and only if element-wise preconditioners $D^{\mathcal{L}}$ and $D^{\mathcal{L}_{new}}$ for losses $\mathcal{L}$ and $\mathcal{L}_{new}$ satisfy*

$$D^{\mathcal{L}_{new}} = A \odot D^{\mathcal{L}}.$$

The theorem shows that invariance is preserved precisely when the preconditioners transform consistently with the underlying reparametrization: $L, R$ under affine changes and $D$ under coordinate-wise scaling. This provides a clear and practical criterion: the property is not abstract, but follows directly from simple structural relations between preconditioners. Using Theorem 3.1 we now easily proof the scale invariance of `MuAdam-SANIA`.

**Corollary 3.2.** `MuAdam-SANIA` *with $\varepsilon = 0$ is scale invariant.*

*Proof.* Under coordinate-wise rescaling $\mathcal{L}(W) \mapsto \mathcal{L}_{\text{new}}(W) = \mathcal{L}(A \odot W)$, the gradients transform as $G^{\mathcal{L}_{\text{new}}} = A \odot G^{\mathcal{L}}$. By induction, the preconditioner update $D_t^{\mathcal{L}} = [\beta_2 D_{t-1}^{\mathcal{L}} + (1 - \beta_2)(G_t^{\mathcal{L}} \odot G_t^{\mathcal{L}})]^{\circ 1/2}$ for the function $\mathcal{L}_{\text{new}}$ transforms as

$$D_t^{\mathcal{L}_{\text{new}}} = [\beta_2 D_{t-1}^{\mathcal{L}_{\text{new}}} + (1 - \beta_2)(G_t^{\mathcal{L}_{\text{new}}} \odot G_t^{\mathcal{L}_{\text{new}}})]^{\circ 1/2} \quad = A \odot [\beta_2 D_{t-1}^{\mathcal{L}} + (1 - \beta_2)(G_t^{\mathcal{L}} \odot G_t^{\mathcal{L}})]^{\circ 1/2}$$

$$= A \odot D_t^{\mathcal{L}} \quad \text{for all } t.$$

Therefore, $D_t$ for `MuAdam-SANIA` satisfies the condition of Theorem 3.1 for scale invariance. $\qquad \square$

## 4 EXPERIMENTS

### 4.1 SCALE INVARIANCE UNDER COORDINATE-WISE RESCALING

We evaluate scale invariance using the Mushrooms dataset from LIBSVM (Chang & Lin, 2011) with a two-layer MLP. To simulate ill-conditioned feature scales, we construct a scaled variant $\tilde{X} = X \operatorname{diag}(e)$ where $e_i = \exp(a_i)$ and $a_i \sim \operatorname{Uniform}[-k, k]$ independently. We compare non-scale-invariant methods (AdamW, Muon) against scale-invariant baselines (Adam-SANIA) and our spectral hybrid (`MuAdam-SANIA`, Algorithm 1 with $p = 1/2$).

Hyperparameters are tuned using Optuna on validation splits (see Appendix C). As shown in Figure 1, AdamW and Muon degrade on scaled data with higher training loss and reduced test accuracy. In contrast, Adam-SANIA and `MuAdam-SANIA` maintain overlapping trajectories between original and scaled settings due to scale invariance. `MuAdam-SANIA` consistently matches or outperforms Adam-SANIA, demonstrating that combining diagonal preconditioning with a spectral LMO yields tangible gains.

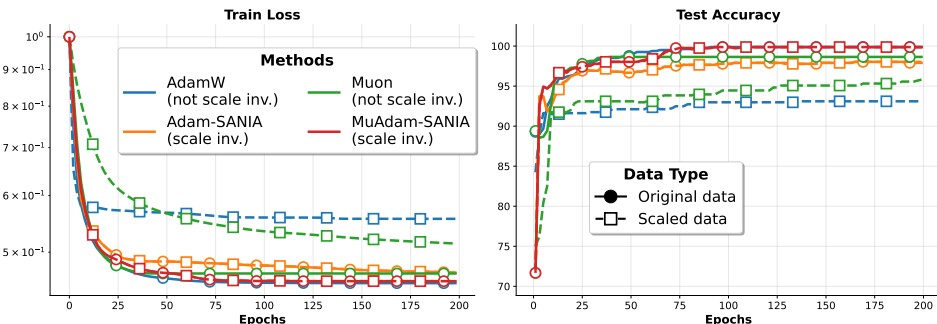

Figure 1: Scale invariance experiment (Mushrooms, LIBSVM) with a two-layer MLP. Training loss (left, log-scale) and test accuracy (right) on original vs. scaled inputs.

### 4.2 GLUE BENCHMARK EVALUATION

We fine-tune DistilBERT base on GLUE tasks (Wang et al., 2018), comparing `AdamW`, `Muon`, and `MuAdam`. We evaluate both LoRA and full fine-tuning, sweeping four learning rates per dataset and tracking validation metrics. Table 2 presents results with columns for datasets and metrics (Matthews correlation for CoLA, accuracy for classification, combined score for STS-B). The `ALL` column shows task averages.

Full fine-tuning yields higher absolute performance than LoRA, while LoRA remains competitive and parameter-efficient. `MuAdam` demonstrates competitive performance, often matching or exceeding baselines across tasks, validating the effectiveness of combining spectral geometry with adaptive preconditioning in transformer fine-tuning.

| | | CoLA Matthews | MNLI Acc | MRPC Acc | QNLI Acc | QQP Acc | RTE Acc | SST-2 Acc | STS-B Comb. | ALL Avg |
|---|---|---|---|---|---|---|---|---|---|---|
| LoRA | AdamW | **0.5152** | 0.7213 | 0.8505 | 0.8572 | **0.8498** | 0.6606 | 0.8979 | 0.8497 | 0.7753 |
| | Muon | 0.5014 | 0.7064 | **0.8676** | 0.8503 | 0.8322 | **0.6679** | 0.8888 | 0.8428 | 0.7697 |
| | MuAdam | 0.5106 | **0.7347** | 0.8627 | **0.8728** | 0.8463 | 0.6643 | **0.9025** | **0.8513** | **0.7806** |
| Full | AdamW | **0.5294** | 0.7824 | **0.8627** | **0.8830** | 0.8780 | 0.6643 | **0.9083** | **0.8625** | **0.7963** |
| | Muon | 0.4963 | 0.6751 | 0.8480 | 0.8378 | 0.8244 | **0.6751** | 0.8945 | 0.8435 | 0.7618 |
| | MuAdam | 0.4979 | **0.7916** | 0.8186 | 0.8827 | **0.8874** | 0.6101 | 0.9014 | 0.8599 | 0.7812 |

Table 2: GLUE (LoRA and Full fine-tuning): datasets are columns with metric under the dataset name; `ALL` is the average over tasks.

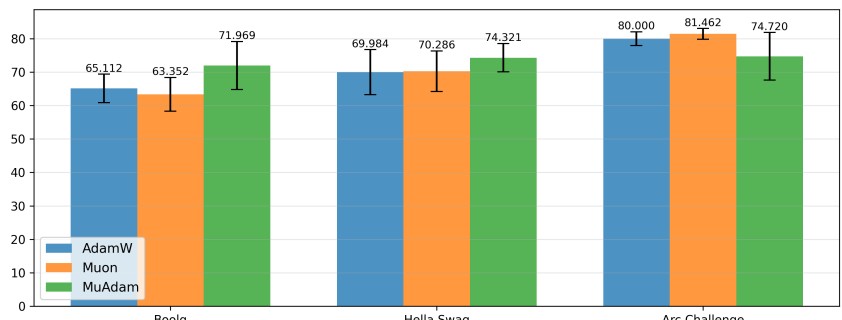

Figure 2: LLM fine-tuning results on Qwen2-7B: mean final accuracy with standard deviation across three seeds.

### 4.3 LLM Fine-Tuning

We fine-tune Qwen2-7B on BoolQ (Clark et al., 2019), HellaSwag (Zellers et al., 2019), and ARC-Challenge (Clark et al., 2018) using LoRA, comparing AdamW, Muon, and `MuAdam` across four learning rates with three random seeds. We report the best accuracy over learning rate sweeps, averaged across seeds with standard deviation error bars.

Figure 2 shows `MuAdam` achieves competitive performance, exceeding both AdamW and Muon on BoolQ and HellaSwag while underperforming on ARC-Challenge, demonstrating effectiveness across question-answering and reasoning tasks.

### 4.4 Character-Level Language Modeling

We evaluate character-level language modeling on the Shakespeare dataset using transformer models with 2, 3, and 4 layers (128 dimensions for 2 layers, 256 for others). Models are trained for 500 epochs with sequence length 256. We perform hyperparameter tuning via random search across batch sizes, learning rates, and dropout values for each optimizer (`AdamW`, `Muon`, `MuAdam`).

Table 3 shows `MuAdam` outperforms `AdamW` on 3 and 4 layer models while slightly underperforming on 2 layers. Both significantly outperform `Muon` across all configurations, validating that our method successfully combines spectral geometry with adaptive preconditioning.

|  | 2 layers Val Acc | 3 layers Val Acc | 4 layers Val Acc |
|---|---|---|---|
| AdamW | **0.5506** | 0.5580 | 0.5636 |
| Muon | 0.5382 | 0.5274 | 0.5568 |
| MuAdam | 0.5483 | **0.5597** | **0.5664** |

Table 3: Shakespeare character-level language modeling: best validation accuracy by optimizer and layer count.

## 5 Conclusion

We introduced a unified framework for optimization with matrix-parameterized models based on preconditioned norms. This abstraction subsumes steepest descent, quasi-Newton, and adaptive methods, and provides the first systematic characterization of affine and scale invariance in the matrix setting. Building upon this foundation, we proposed `MuAdam` and `MuAdam-SANIA`, which combine spectral geometry with adaptive preconditioning, achieving strong empirical results across diverse tasks. These results suggest that integrating generalized norms with structured preconditioning offers a rich and still underexplored landscape for the development of next-generation optimization methods.

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

Table 4: Table of the frequently used notation.

| Notation | Meaning |
|---|---|
| $\mathcal{L}$ | Objective loss |
| $\otimes$ | Kronecker product |
| $\odot$ | Hadamard product |
| $A^{-T}$ | Inverse transpose (of an invertable matrix) |
| $A^{\circ-1}$ | Elementwise inverse of the matrix A. |
| $W_t$ | Iterate sequence |
| $\Delta W_t$ | Update rule in the interation $t$ |
| $G_t$ | Gradient at iterate $t$ |
| $L_t, R_t, D_t, V_t$ | Left, right, diagonal, elementwise preconditioners |
| $H_t^L, H_t^R$ | Left and right curvature factors |
| $\|\cdot\|_\alpha, \|\cdot\|_\beta$ | Vector norms |
| $\|\cdot\|_F$ | Frobenius norm |
| $\|\cdot\|_{\mathrm{RMS}}$ | RMS norm $= \dim(x)^{\frac{1}{2}}\|x\|_2$ |
| $\dim(x)$ | dimension of vector x |
| $\|\cdot\|_{\alpha\to\beta}$ | Matrix norm induced by vector norms $\|\cdot\|_\alpha, \|\cdot\|_\beta$ |
| $(L, R)$-norm, $D$-norm | Preconditioned matrix norm |
| $lmo(\cdot), lmo_{\|\cdot\|}(\cdot), lmo_{L,R,\|\cdot\|}(\cdot), lmo_{D,\|\cdot\|}(\cdot)$ | Linear minimization oracles with dependence on the base norm and preconditioners |

## A  ADDITIONAL DISCUSSION ON GEOMETRIC INVARIANCE

In this subsection we collect the algebraic details that justify the summary given in Section 3. Throughout we fix an invertible matrix $A \in \mathbb{R}^{d\times d}$ and consider the re–parameterized loss for vectorized parameters

$$\Phi(w^A) := \mathcal{L}\big(A\,w^A\big), \qquad w^A := A^{-1}w,$$

here we used $\Phi$ instead of $\mathcal{L}_{\mathrm{new}}$ as in Section 3 in terms of convenience, we will do the similar change of notation in the Appendix B. If an optimizer produces iterates $w_t$ for $\mathcal{L}$, we denote by $w_t^A$ the iterates it produces on $\Phi$. The method is *affine–invariant* iff $w_t^A = A^{-1}w_t$ for all $t$. A weaker property is obtained when $A$ is restricted to be diagonal with positive entries; this is called *scale invariance*.

**Newton's method (Nesterov et al., 2018) (affine invariant).**  With the Hessian $H_t = \nabla^2\mathcal{L}(w_t)$, one step of Newton reads

$$w_{t+1} = w_t - H_t^{-1}\nabla\mathcal{L}(w_t).$$

Under the change of variables we have $\nabla\Phi(w_t^A) = A^\top\nabla\mathcal{L}(w_t)$ and $\nabla^2\Phi(w_t^A) = A^\top H_t A$. Hence

$$w_{t+1}^A = w_t^A - \big(A^\top H_t A\big)^{-1}A^\top\nabla\mathcal{L}(w_t) = A^{-1}\big(w_t - H_t^{-1}\nabla\mathcal{L}(w_t)\big) = A^{-1}w_{t+1},$$

therefore Newton's iterates transform equivariantly and the method is fully affine invariant.

**Stochastic gradient descent (Robbins & Monro, 1951) (not invariant).**  SGD updates are

$$w_{t+1} = w_t - \gamma_t\,g_t, \qquad g_t := \nabla\mathcal{L}(w_t, \xi_t).$$

After re–parameterisation,

$$w_{t+1}^A = A^{-1}w_t - \gamma_t A\,g_t \neq A^{-1}w_{t+1},$$

therefore neither affine nor scale invariance is satisfied.

**Adam (Kingma & Ba, 2014) (with $\varepsilon = 0$) (not scale invariant).**  With exponential moving averages $m_t, v_t$ the Adam step is

$$w_{t+1} = w_t - \gamma_t\,\frac{m_t}{\sqrt{v_t}}, \quad m_t = \beta_1 m_{t-1} + (1-\beta_1)g_t, \quad v_t = \beta_2 v_{t-1} + (1-\beta_2)g_t \odot g_t.$$

Under a diagonal rescaling $A = \mathrm{diag}(a_1, \ldots, a_d)$ (coordinate–wise scale change) we have $m_t^A = Am_t$ and $v_t^A = A^2 v_t$, so

$$w_{t+1}^A = A^{-1} w_t - \gamma_t A \frac{m_t}{\sqrt{v_t}} \neq A^{-1} w_{t+1}.$$

The mismatch comes from the square root in the denominator.

**SANIA (Abdukhakimov et al., 2023) (scale invariant).** SANIA removes the square root and normalises by $v_t$ itself:

$$w_{t+1} = w_t - \gamma_t \frac{m_t}{v_t}.$$

With the same relations $m_t^A = Am_t$ and $v_t^A = A^2 v_t$ we obtain

$$w_{t+1}^A = A^{-1} w_t - \gamma_t A^{-1} \frac{m_t}{v_t} = A^{-1} w_{t+1},$$

which shows exact scale invariance. (Full affine invariance is not expected because the pre–conditioner is diagonal.)

## B  MISSING PROOFS

### B.1  PROOF OF THEOREM 2.3

*Proof.* By definition of the linear minimization oracle (4), we have for a general preconditioned norm

$$\mathrm{lmo}_{\mathcal{P}, \|\cdot\|}(G) = \arg\min_{T: \|\mathcal{P}(T)\| \leq \rho} \langle G, T \rangle,$$

where $\mathcal{P}$ denotes the preconditioning transform. We now proceed case by case.

(i) $(L, R)$-norm (Definition 2.1). Here $\mathcal{P}(T) = LTR$. Setting $Q = LTR$ (so $T = L^{-1}QR^{-1}$), and noting that the mapping is bijective since $L, R$ are non-degenerate, we obtain

$$\begin{aligned}
\mathrm{lmo}_{L, R, \|\cdot\|}(G) &= L^{-1} \cdot \arg\min_{\|Q\| \leq \rho} \langle G, L^{-1}QR^{-1} \rangle \cdot R^{-1} \\
&= L^{-1} \cdot \arg\min_{\|Q\| \leq \rho} \langle L^{-T}GR^{-T}, Q \rangle \cdot R^{-1} \\
&= L^{-1} \cdot \mathrm{lmo}_{\|\cdot\|}(L^{-T}GR^{-T}) \cdot R^{-1}.
\end{aligned}$$

(ii) $D$-norm (Definition 2.2). Here $\mathcal{P}(T) = D \odot T$. Let $Q = D \odot T$, so that $T = D^{\circ-1} \odot Q$. Since $D$ has strictly positive entries, this mapping is also bijective. Substituting, we get

$$\begin{aligned}
\mathrm{lmo}_{D, \|\cdot\|}(G) &= D^{\circ-1} \odot \arg\min_{\|Q\| \leq \rho} \langle G, D^{\circ-1} \odot Q \rangle \\
&= D^{\circ-1} \odot \arg\min_{\|Q\| \leq \rho} \langle D^{\circ-1} \odot G, Q \rangle \\
&= D^{\circ-1} \odot \mathrm{lmo}_{\|\cdot\|}(D^{\circ-1} \odot G).
\end{aligned}$$

Thus, in both cases the LMO acts not before or after preconditioning, but within a transformed gradient space, yielding the claimed formulas for $(L, R)$- and $D$-norms. $\square$

### B.2  PROOF OF THEOREM 3.1

First we need to prove a more general theorem about affine invariance and arbitrary norm. Also, as in Appendix A, for convenience in this section, we will use a notation $\Phi$ for the changed function, instead of $\mathcal{L}_{\mathrm{new}}$ as we did it in Section 3.

**Theorem B.1.** *Define norms we use for running algorithm* (5) *for functions* $\mathcal{L}(W)$ *and* $\phi(\Theta)$ *respectively as* $\|\cdot\|_{\mathcal{L}}$ *and* $\|\cdot\|_{\phi}$. *Then for the step* (5) *to be affine invariant it is necessary that for all matrices* $T$ *of the proper shape and for all nondegenerate matrices* $A_L$ *and* $A_R$ *of the proper shape to satisfy this equation:*

$$\|T\|_{\mathcal{L}} = \left\|A_L^{-1} T A_R^{-1}\right\|_{\phi}.$$

*In order for this condition to become sufficient, we need to require the uniqueness of finding* $\arg\min$ *in lmo* (4), *i.e., for all* $G$ *of the proper shape it holds that*

$$\left|\arg \min_{\|T\|_{\mathcal{L}} \leq \rho} \{\langle G, T\rangle\}\right| = 1.$$

*Proof.* Since $\phi(\Theta) = \mathcal{L}(A_L \Theta A_R)$, for any optimization algorithm to be affine invariant it is necessary and sufficient to it's output to satisfy $\Theta^t = A_L^{-1} W_t A_R^{-1}$. Since steps in the steepest descent (5) are of the form

$$W^{t+1} = W^t - \text{lmo}_{\mathcal{L}}(\nabla \mathcal{L}(W_t)) \ \text{ and } \ \Theta^{t+1} = \Theta^t - \text{lmo}_{\phi}(\nabla \phi(\Theta_t)),$$

where $\text{lmo}_{\mathcal{L}}$ and $\text{lmo}_{\phi}$ are linear minimization oracles based on the norms $\|\cdot\|_{\mathcal{L}}$ and $\|\cdot\|_{\phi}$.

From the mathematical induction and the fact that $\Theta^0 = A_L^{-1} W_0 A_R^{-1}$, condition $\Theta^t = A_L^{-1} W_t A_R^{-1}$ equivalent to:

$$\text{lmo}_{\phi}(\nabla \phi(\Theta_t)) = A_L^{-1} \cdot \text{lmo}_{\mathcal{L}}(\nabla \mathcal{L}(W_t) \cdot A_R^{-1}. \tag{8}$$

For the function $\mathcal{L}(W)$ we have

$$\text{lmo}_{\mathcal{L}}(\nabla \mathcal{L}(W_t)) = \arg \min_{\|Q\|_{\mathcal{L}} \leq \rho} \{\langle \nabla \mathcal{L}(W_t), Q\rangle\}. \tag{9}$$

For the function $\phi(\Theta)$, if $\Theta^t = A_L^{-1} W_t A_R^{-1}$, we have

$$\nabla \phi(\Theta_t) = \nabla_{\Theta_t} \mathcal{L}(A_L \Theta A_R) = A_L^T \nabla_{A_L \Theta A_R} \mathcal{L}(A_L \Theta_t A_R) A_R^T = A_L^T \nabla \mathcal{L}(W_t) A_R^T.$$

Therefore lmo for the function $\phi$ takes form

$$\text{lmo}_{\phi}(\nabla \phi(\Theta_t)) = \text{lmo}_{\phi}(A_L^T \nabla \mathcal{L}(W_t) A_R^T) = \arg \min_{\|Q\|_{\phi} \leq \rho} \left\{\langle A_L^T \nabla \mathcal{L}(W_t) A_R^T, Q\rangle\right\}$$

$$= \arg \min_{\|Q\|_{\phi} \leq \rho} \{\langle \nabla \mathcal{L}(W_t), A_L Q A_R\rangle\}$$

Since matrix $A_{L,R}$ are nondegenerate, we can make a variable substitution $T = A_L Q A_R$ and obtain that

$$\text{lmo}_{\phi}(\nabla \phi(\Theta_t)) = A_L^{-1} \cdot \arg \min_{T: \left\|A_L^{-1} T A_R^{-1}\right\|_{\phi} \leq \rho} \{\langle \nabla \mathcal{L}(W_t), T\rangle\} \cdot A_R^{-1} \tag{10}$$

Combining equations (8), (9) and (10), we can obtain that for the affine invariance it is necessary and sufficient that for all matrixes $G$ of the proper shape, this equality holds (we change $Q$ to $T$ in (9) for convenience):

$$\arg \min_{\|T\|_{\mathcal{L}} \leq \rho} \{\langle G, T\rangle\} = \arg \min_{T: \left\|A_L^{-1} T A_R^{-1}\right\|_{\phi} \leq \rho} \{\langle G, T\rangle\}.$$

Since minimization functions are the same for both $\arg\min$, the necessary condition of affine invariance is that for all matrices of the proper shape:

$$\|T\|_{\mathcal{L}} = \left\|A_L^{-1} T A_R^{-1}\right\|_{\phi} \tag{11}$$

However equation (11) is not sufficient, because we need also to require the uniqueness of this $\arg\min$. $\qquad\square$

We now ready to proof Theorem 3.1.

*Proof Theorem 3.1.* The proof of Theorem 3.1 consists of a straightforward application of Theorem B.1. For all matrices $T \in \mathbb{R}^{m \times n}$ the following equality should hold:

$$\|L_{\mathcal{L}} T R_{\mathcal{L}}\| = \|L_{\phi} A_L^{-1} T A_R^{-1} R_{\phi}\|.$$

Therefore the necessary condition on the affine invariance is

$$L_{\mathcal{L}} = L_{\phi} A_L^{-1} \text{ and } R_{\mathcal{L}} = A_R^{-1} R_{\phi}.$$

In order for this condition to become sufficient, we need to require the uniqueness of finding $\arg\min$ in lmo with the norm $\|\cdot\|_{L,R,\|\cdot\|}$. Since matrices $L$ and $R$ for $\mathcal{L}(W)$ and $\phi(\Theta)$ are non-degenerative, the uniqueness deepens only on the norm $\|\cdot\|$, i.e.,

$$\left| \arg\min_{\|T\| \leq \rho} \{\langle G, T \rangle\} \right| = 1.$$

The scale invariance case is proven in a similar way as for Theorem 2.3. □

## C  SCALE INVARIANCE SETUP AND HYPERPARAMETERS

To ensure a fair comparison across optimizers and input scalings, we perform hyperparameter tuning separately for each method and for both the original and scaled tasks using Optuna on a held-out validation split (see Section 4.1). Tuned values are selected to maximize validation accuracy, and final results are reported on the test split with the chosen configuration. Below we summarize the key training and tuning settings used in our experiments.

Table 5: Summary of training and tuning hyperparameters.

| Parameter | Value |
|---|---|
| Learning rate (tuned by Optuna) | LogUniform$[1e\text{-}6, 5e0]$ |
| Weight decay (tuned by Optuna) | LogUniform$[1e\text{-}6, 1e\text{-}2]$ |
| Batch size | `len(train_dataloader)` (full-batch) |
| Training epochs | 200 |
| Tuning epochs | 10 |
| Optuna runs per setting | 40 |
| Hidden dimension (MLP) | 100 |
| Bias terms | Disabled |
| Weight initialization | Input layer: zeros; Output layer: uniform |
| Numerical epsilon | $1e{-}40$ |
| DType | float64 |
| Scaling bound for data | $k = 10$ |
| Seeds | $\{18, 52, 812\}$ |
| Optimizers compared | AdamW, Muon, Adam-SANIA, `MuAdam-SANIA` |

Notes: The diagonal preconditioner $D_t$ includes an additive $\varepsilon$, which normally takes values around $10^{-8}$ (Kingma & Ba, 2014), however this value is big enough to break down scale invariance. To minimize this effect we use $\varepsilon = 10^{-40}$, therefore its contribution is negligible while still ensuring numerical safety. In addition, we employ float64 precision, since the use of such a tiny $\varepsilon$ further increases the need for high numerical accuracy, and scale invariance requires well-defined updates even under extremely large or small entries of the scaling matrix $A$.

Also, the input layer of the MLP is initialized with zeroes, which is consistent with the scale-invariance assumption $W_0^{\text{new}} = A^{-1} W_0 = 0$. At the same time, the final output layer is initialized with a standard uniform distribution, since setting it to zero would eliminate the signal necessary for learning and prevent the network from training.

Table 6: GLUE LoRA fine-tuning: training setup and sweep.

| Parameter | Value |
|-----------|-------|
| Backbone | `distilbert/distilbert-base-uncased` |
| Fine-tuning | LoRA ($r{=}4$, $\alpha{=}32$, dropout 0.05) |
| Batch size | 16 |
| Grad. accumulation | 2 |
| DType | bfloat16 |
| LR scheduler | linear, warmup ratio 0.1 |
| Max train steps | 10000 |
| Eval/save | eval each epoch; no checkpoint saving |
| Sweep LRs | $\{2{\times}10^{-4},\ 10^{-4},\ 5{\times}10^{-5},\ 3{\times}10^{-5}\}$ |
| Optimizers compared | AdamW, Muon, `MuAdam` |

Notes: results are picked as the best validation score observed across all evaluation steps within each run, then the best over the LR sweep per optimizer.

## D  LLM FINE-TUNING SETUP AND HYPERPARAMETERS

We fine-tune Qwen2-7B on three reasoning datasets using LoRA with a grid search over learning rates. Results are averaged over multiple seeds for statistical robustness.

Table 7: LLM fine-tuning: training setup and sweep.

| Parameter | Value |
|-----------|-------|
| Backbone | `Qwen/Qwen2-7B` |
| Fine-tuning | LoRA ($r{=}16$, $\alpha{=}32$, dropout 0.05) |
| Batch size | 1 |
| Grad. accumulation | 4 |
| Quantization | 4-bit |
| DType | bfloat16 |
| LR scheduler | linear, warmup ratio 0.1 |
| Max train steps | 1000 |
| Max seq length | 512 |
| Datasets | BoolQ, HellaSwag, ARC-Challenge |
| Sweep LRs | $\{2{\times}10^{-4},\ 10^{-4},\ 5{\times}10^{-5},\ 3{\times}10^{-5}\}$ |
| Seeds | $\{42, 123, 456\}$ |
| Optimizers compared | AdamW, Muon, `MuAdam` |

Notes: final accuracy is selected as the best value across all evaluation steps within each run, then the best over the LR sweep per optimizer. Results are averaged across three seeds with standard deviation reported.

# E CHARACTER-LEVEL LANGUAGE MODELING SETUP AND HYPERPARAMETERS

We evaluate optimizers on character-level language modeling using the Shakespeare dataset with transformer models of varying architecture complexity. Hyperparameters are tuned using random search across multiple configurations to ensure fair comparison between optimizers.

Table 8: Shakespeare character-level language modeling: training setup and sweep.

| Parameter | Value |
|---|---|
| Model | Transformer (base config) |
| Dataset | `shakespeare-char` |
| Model layers | 2, 3, 4 |
| Attention heads | 4 |
| Embedding dim | 128 (2 layers), 256 (3-4 layers) |
| Vocab size | 96 |
| Batch size | Random search: {32, 128, 256} |
| Sequence length | 256 |
| Grad clip | 0.5 |
| Weight decay | 0.1 |
| LR scheduler | Cosine |
| Dropout | Random search: {0.05, 0.1, 0.15, 0.25} |
| Beta1, Beta2 | 0.9, 0.999 |
| LR sweep | $\{10^{-6}, 5 \cdot 10^{-6}, 10^{-5}, 5 \cdot 10^{-5}, 10^{-4}, 5 \cdot 10^{-4}, 10^{-3}, 5 \cdot 10^{-3}, 10^{-2}, 3 \cdot 10^{-2}\}$ |
| Optimizers compared | AdamW, Muon, `MuAdam` |

# F THE USE OF LARGE LANGUAGE MODELS (LLMS)

We use Large Language Models for text editing, i.e. grammar checking, word selection, text compression and coding/visualization.

