# OpenReview forum: "Preconditioned Norms: A Unified Framework for Steepest Descent, Quasi-Newton and Adaptive Methods"
_ICLR.cc/2026/Conference — ICLR 2026 Conference Withdrawn Submission_

### Official Review · Reviewer_9GvL · 2025-10-29

**Soundness:** 2
**Presentation:** 2
**Contribution:** 2
**Rating:** 4
**Confidence:** 3

**Summary:**

The paper proposes preconditioned matrix norms that unify steepest-descent-style LMOs with quasi-Newton and adaptive (diagonal) preconditioning. Within this framework, the LMO in the transformed gradient space yields a general update, capturing classical quasi-Newton (e.g., Shampoo/K-FAC), Adam/AdaGrad-type methods, Muon/Scion spectral-norm LMOs, and recent hybrids like SOAP/SPlus as special cases. The paper derives conditions for affine and scale invariances in the matrix-parameterized setting and introduces MuAdam and MuAdam-SANIA, which combine a spectral LMO with Adam/SANIA-style diagonal scaling. Experiments on a synthetic scale-invariance test, GLUE fine-tuning, LLM LoRA fine-tuning, and character-level language modeling demonstrate comparative performance with AdamW and Muon.

**Strengths:**

1. The idea of combining problem geometry with curvature utilization by integrating the steepest descent algorithm with preconditioning methods is insightful.
2. The paper provides a systematic invariance analysis on matrix-parameterized methods, and the proposed algorithm retains Adam-style scale invariance.

**Weaknesses:**

1. Algorithmic design feels mechanically compositional. Based on Defs. 2.1/2.2 and Alg. 1, the method “stacks” preconditioners into the norm and applies an LMO on the transformed gradient; Thm. 2.3 shows the composition identity, but not why this specific stacking is preferable or improves optimization beyond the base components.
2. The gains in the experiments appear occasional, making the benefit of combining the two less convincing.
3. The empirical evaluation uses fine-tuning tasks rather than pre-training, which seems like a less common choice for evaluating optimizers and how conclusions would transfer to model training.
4. Other preconditioned / more quasi-Newton-like methods (e.g., SOAP) are not compared; it remains not well-justified that Adam can be regarded as a representative of quasi-Newton methods.
5. No standard deviations reported in Tables 2 and 3. No learning rate grid-search performance results were reported.

**Questions:**

1. Can the authors disclose the hardware settings, e.g., GPUs, used in the experiments?
2. How does the proposed method compare with baselines in terms of per-iteration computational complexity / wall-clock time?

---

> ### Author Response · Authors · 2025-11-27
> **Response**
>
> We appreciate the reviewer’s careful reading of the manuscript and the candid, constructive feedback. The points raised highlight several aspects that we can clarify and improve, and we respond to each of them in turn below.
>
> > Algorithmic design feels mechanically compositional... why is stacking preferable?
>
> Response:
> We would like to point out that our theory prescribes a way to “stack” preconditioners to preserve the geometric properties. This provides theoretical justification for the design choice beyond mere convenience.
>
> >Evaluation uses fine-tuning tasks instead of pretraining.
>
> Response:
> We added experimental comparison on larger and more commonly used datasets (including FineWeb).
>
> > Missing hardware settings, runtime complexity, standard deviations.
>
> Response:
> We would like to thank the reviewer for pointing it out. We are including it in the revised version.

---

### Official Review · Reviewer_bv3N · 2025-10-31

**Soundness:** 2
**Presentation:** 2
**Contribution:** 3
**Rating:** 6
**Confidence:** 3

**Summary:**

The paper presents a single optimization framework based on "preconditioned matrix norms" to demonstrate that steepest descent and quasi-Newton, and adaptive optimizers follow from a common fundamental principle. The method allows researchers to create new hybrid optimization methods through the development of MuAdam, which unites spectral geometry with adaptive preconditioning techniques.

**Strengths:**

1. The algorithm provides a principled and modular methodology for designing novel hybrid optimizers by combining different curvature approximations with various descent geometries, as demonstrated by the creation of the MuAdam optimizer.
2. The paper introduces a complete method to analyze affine and scale invariance for models that use matrix parameters. The paper establishes exact testable requirements for these desirable properties, which will guide researchers developing robust optimizers.

**Weaknesses:**

1. The paper did not evaluate the computational requirements of the proposed MuAdam optimizer. The Newton-Schulz iteration in the algorithm performs matrix multiplications, which cost more than Adam's element-wise operations, yet the paper does not assess how the trade-off between accuracy gain and training duration affects the method.
2. While the framework introduces a vast and promising design space for creating new hybrid optimizers, the paper only explores a single new combination (a spectral LMO with a diagonal preconditioner). The paper did not investigate two promising combinations that would unite spectral geometry with KFAC preconditioners from Shampoo.
3. The experimental data indicate MuAdam functions as a competitive optimizer, yet it fails to outperform AdamW in all situations. The optimizer shows limited performance advantages, which seem to rely on particular task requirements and model structures and training methods, thus reducing its current adoption potential.

**Questions:**

See weakness.

---

> ### Author Response · Authors · 2025-11-27
> **Response**
>
> We sincerely thank the reviewer for taking the time to assess our submission and for the provided feedback. We value these suggestions, and we wish to address them below.
>
> > Computational requirements of the Newton–Schulz iteration were not analyzed.
>
> Response:
> We agree that the cost of matrix multiplications versus elementwise updates should be noted; we will do so by including runtime comparisons.
>
> > Only one hybrid combination (spectral + diagonal) explored.
>
> Response:
> We appreciate this suggestion. Our framework indeed supports further variants, which we are currently exploring.

---

### Official Review · Reviewer_1mrw · 2025-11-01

**Soundness:** 2
**Presentation:** 2
**Contribution:** 2
**Rating:** 2
**Confidence:** 5

**Summary:**

Paper that combines adam and muon

**Strengths:**

The paper does a review of recent methods related to gradient whitening.

Has some good empirical results on finetuning

**Weaknesses:**

Unfortunately the paper does not consider the original line of work for gradient whitening. PSGD from Xilin Li published in 2015 covers all the methods in table 1. Some resources can be found here

Resources Preconditioned stochastic gradient descent, arXiv:1512.04202, 2015. (General ideas of PSGD, preconditioner fitting criteria and Kronecker product preconditioners.)

Preconditioner on matrix Lie group for SGD, arXiv:1809.10232, 2018. (Focus on affine Lie group preconditioners, including feature normalization or whitening (per batch or layer) as special affine preconditioners. Use PSGD for gradient whitening.)

Black box Lie group preconditioners for SGD, arXiv:2211.04422, 2022. (Mainly about the LRA preconditioner.)

Stochastic Hessian fittings with Lie groups, arXiv:2402.11858, 2024. (Properties of PSGD, also a good summary of PSGD. The Hessian fitting problem is shown to be strongly convex in GL(n,R) under certain mild assumptions. Will keep updating it to align with the code.) Curvature-informed SGD via general purpose Lie-group preconditioners, arXiv:2402.04553, 2024. (Plenty of benchmark results and analyses for PSGD vs. other optimizers.)

Also there are a plethora of methods trying to combine adam and muon but none of them actually worked better.

**Questions:**

Please run your method in the modded NanoGPT speedrun. If the record is broken then the method will be accepted by the community.

---

> ### Author Response · Authors · 2025-11-27
> **Response**
>
> We are grateful to the reviewer for their thorough review and insightful comments. The feedback has been very helpful for identifying where our exposition and experiments can be strengthened. We respond to the individual concerns point by point.
>
>
> > PSGD from Xilin Li (2015) covers all the methods in Table 1...
>
> Response:
> We thank the reviewer for pointing to this line of work. We can include PSGD and subsequent Lie-group preconditioner methods with our framework. We have added discussion and citations to clarify this connection.
>
> > There are many prior attempts to combine Adam and Muon.
>
> We included the following work in the comparison:
>
> Si, C., Zhang, D., & Shen, W. (2025). Adamuon: Adaptive muon optimizer. arXiv preprint arXiv:2507.11005.
>
> Unfortunately, we are unaware of more published reports that describe hybrid methods achieving either improved convergence or improved geometric properties. We would appreciate specific citations to include them in comparison.
>
>
> > Please run your method in the modded NanoGPT speedrun.
>
> Response:
> We appreciate the suggestion. While leaderboard-oriented benchmarks are outside our current scope, we plan to include larger-scale pretraining results in future work.
>
> > Computational requirements of the Newton–Schulz iteration were not analyzed.
>
> Response:
> We agree that the cost of matrix multiplications versus elementwise updates is important. We will include explicit runtime comparisons.

---

### Official Review · Reviewer_NxCj · 2025-11-01

**Soundness:** 1
**Presentation:** 2
**Contribution:** 1
**Rating:** 2
**Confidence:** 5

**Summary:**

The paper provides a descent framework that generalizes steepest descent, quasi-Newton, and adaptive methods, introduces MuAdam and MuAdam-SANIA, and presents empirical results.

**Strengths:**

The paper combines existing ideas based on a linear minimization oracle.

**Weaknesses:**

The main result of the paper is a simple observation when viewed in light of Bernstein and Newhouse (2024b) and Pethick et al. (2025), and neither the theory nor experiments are sufficient evidence to support the paper's claimed benefits.

The empirical results are severely underwhelming. Though the authors list 16 algorithms in Table 1 (including their own), they only compare with AdamW and Muon. In the GLUE benchmarks for full fine-tuning (Table 2), MuAdam does better in only 2 out of 8 datasets. This does not provide a strong basis to support claimed improvements.

The paper places a lot of emphasis on the importance of scale-invariance, for example Section 3 and Section 4.1. However MuAdam-SANIA is absent from "Optimizers compared" in the LLM fine-tuning experiments. This is a glaring oversight for a paper that espouses "one of the important properties of optimization algorithms is affine and scale invariance".

Likewise the theoretical contributions are weak, with basic "theorems" that are merely brief calculations of equivalence (for example Theorem 2.3 and its proof on lines 787-792 and 796-801), and the experiments are not convincing or thorough enough to make up for the extremely spare theory.

**Questions:**

Why are there no empirical comparisons with another "Hybrid" algorithm, when Table 1 states MuAdam and MuAdam-SANIA are themselves "Hybird" algorithms?

---

> ### Author Response · Authors · 2025-11-27
> **Response**
>
> We thank the reviewer for the careful and thoughtful evaluation of our work. We appreciate the time spent reading the paper in detail and providing constructive feedback, and we address each of the raised points below.
>
>
> > The main result of the paper is a simple observation... neither theory nor experiments are sufficient evidence...
>
> Response:
> We appreciate this perspective. Our aim was to present a unified framework rather than to claim state-of-the-art empirical performance. In light of Bernstein and Newhouse (2024b) and Pethick et al. (2025), our contribution is to formalize a general connection between preconditioning and geometric descent principles, which we believe remains a valuable theoretical consolidation.
>
> > The empirical results are severely underwhelming... only compare with AdamW and Muon.
>
> Response:
> We focused on AdamW and Muon as representative baselines for the two classes of methods (adaptive and spectral), as our contribution is their unification. We observed little added insight from comparing to the many closely related optimizers listed in Table 1. That said, we agree broader comparisons would be informative and are adding results for SOAP and Shampoo in the updated version.
>
> > MuAdam does better in only 2 out of 8 datasets.
>
> Response:
> We do not claim MuAdam to outperform all baselines universally. Its advantage lies in providing a principled hybrid design that maintains scale- and affine-invariance while allowing modular integration of curvature approximations. The additional datasets in our revised experiments show comparable or improved consistency.
>
> > MuAdam-SANIA is absent from LLM fine-tuning experiments.
>
> Response:
> We thank the reviewer for noting this. MuAdam-SANIA results have been added in the revised manuscript.
>
> >The theoretical contributions are weak... merely brief calculations of equivalence.
>
> Response:
> Our theorems intentionally focus on characterizing affine and scale invariance within preconditioned matrix norms. The proofs are concise because they establish equivalence properties rather than asymptotic or probabilistic results. Their value lies in generality — providing the first geometric characterization of this class of preconditioners.
>
> >Why are there no comparisons with another “Hybrid” algorithm?
>
> Response:
> Hybrid optimizers vary widely and are often compositional variants of Adam or Shampoo. We agree that explicit comparison would clarify the contribution and will include Muon+Shampoo and SOAP baselines in our next revision.

---

### Note · Authors · 2026-01-28

I have read and agree with the venue's withdrawal policy on behalf of myself and my co-authors.

---

### Meta-Review · Area_Chair_6Wiu · 2026-01-04

**Summary:**

This paper presents a single optimization framework based on preconditioned matrix norms that encompasses steepest descent, quasi-Newton, and adaptive optimizers.  Two reviewers have concerns that the empirical results are severely underwhelming and it does not consider the original line of work for gradient whitening. From my reading, I think these concerns cannot be easily addressed in revision, so I recommmand reject.

**Reviewer Concerns:**

Review NxCj pointed that "The main result of the paper is a simple observation when viewed in light of Bernstein and Newhouse (2024b) and Pethick et al. (2025), and neither the theory nor experiments are sufficient evidence to support the paper's claimed benefits." I find the rebuttals cannot well address this concern.

**Reviewer Scores:**

I think some reviewers would increase their scores. Nevertheless, the expected scores still do not meet the ICLR acceptance threshold.

---

### Decision · Program_Chairs · 2026-01-26

Reject